# Pharmacological Properties of Polyphenols: Bioavailability, Mechanisms of Action, and Biological Effects in In Vitro Studies, Animal Models, and Humans

**DOI:** 10.3390/biomedicines9081074

**Published:** 2021-08-23

**Authors:** Kristine Stromsnes, Rudite Lagzdina, Gloria Olaso-Gonzalez, Lucia Gimeno-Mallench, Juan Gambini

**Affiliations:** 1Freshage Research Group, Department of Physiology, Faculty of Medicine, Institute of Health Research-INCLIVA, University of Valencia and CIBERFES, Avda. Blasco Ibañez, 15, 46010 Valencia, Spain; krisbaks@alumni.uv.es (K.S.); gloria.olaso@uv.es (G.O.-G.); 2Faculty of Medicine, Department of Human Physiology and Biochemistry, Riga Stradins University, LV-1007 Riga, Latvia; Rudite.Lagzdina@rsu.lv; 3Department of Biomedical Sciences, Faculty of Health Sciences, Cardenal Herrera CEU University, 46115 Valencia, Spain; lucia.gimenomallench@uchceu.es

**Keywords:** bioavailability, pharmacological, mechanisms of action, biological effects, polyphenol, in vitro, in vivo, human, concomitant, drugs

## Abstract

Drugs are bioactive compounds originally discovered from chemical structures present in both the plant and animal kingdoms. These have the ability to interact with molecules found in our body, blocking them, activating them, or increasing or decreasing their levels. Their actions have allowed us to cure diseases and improve our state of health, which has led us to increase the longevity of our species. Among the molecules with pharmacological activity produced by plants are the polyphenols. These, due to their molecular structure, as drugs, also have the ability to interact with molecules in our body, presenting various pharmacological properties. In addition, these compounds are found in multiple foods in our diet. In this review, we focused on discussing the bioavailability of these compounds when we ingested them through diet and the specific mechanisms of action of polyphenols, focusing on studies carried out in vitro, in animals and in humans over the last five years. Knowing which foods have these pharmacological activities could allow us to prevent and aid as concomitant treatment against various pathologies.

## 1. Introduction

Polyphenols are a group of chemical substances primarily synthesized in plants as secondary metabolites that are structurally characterized for having one or more hydroxyl groups (–OH) bound with phenols. All of them are indispensable for physiological functions of vegetables and participate in the defense against stresses such as hydric and light stress.

They are abundant in nature and extremely diverse in their molecular structure. They can be found in fruits, leaves, stems, roots, and seeds and to this day, more than 8000 polyphenols have been identified [1]. Therefore, terminology and classification are complex processes. The chemical structures can be divided based on some differences in phenolic acids, stilbenes, phenolic alcohols, lignans, and flavonoids [2] (Figure 1). Furthermore, in nature they can be found in their free form (aglycones), bound to other compounds as sugars (glycosides), or even bound to the extracellular matrix or cell wall.

The major dietary sources are fruits and drinks such as juices, wine, tea, coffee, hot chocolate, and beer, and in lower quantities in vegetables, legumes, and cereals. The total ingestion of polyphenols in a standard diet is approximately 1 g/day. Of this, phenolic acids make up one third of the total, flavonoids make up two thirds, and other variants are ingested in minor quantities [3].

Studies about bioavailability in humans are still scarce. The concentration in plasma is rarely found to be above 1 µM after the consumption of 10 to 100 mg of a single phenolic compound. However, the total phenol concentration in plasma can be augmented depending on the presence of metabolites formed in the tissue of the organism or the microflora in the colon. The chemical structure of a polyphenol determines its absorption rate, degree, and nature of the metabolites that circle in plasma. Therefore, a better understanding of the absorption and metabolites will help us better understand the bioavailability [3] as they possess a large number of pharmacological properties.

The health benefits of polyphenols present in our diet has spiked interest among researchers in the last few years [4,5,6]. Therefore, the objective of this review is to elucidate the bioavailability, pharmacological properties, and pathways that polyphenols act upon to enable us to use these compounds as concomitant treatments or even as prophylactics of multiple pathologies.

## 2. Absorption and Metabolism 

Due to the molecular structure of polyphenols, their absorption is generally high in free form thanks to their lipophilic properties, which allow them to pass through plasmatic membranes by simple diffusion [5]. However, when they form complexes with other compounds, their absorption changes.

Polyphenols in their free form and in complexes can present changes in the structure of the digestive tract including on a hepatic level. These changes are observed from the liberation of the polyphenol, to its union with complex compounds, to the addition of conjugates, and the generation of metabolites [7]. Additionally, the joint ingestion of other compounds with polyphenols can affect its absorption: fibers, high fat diets, and the consumption of alcoholic beverages have an especially high influence. Finally, as will be discussed in the next sections of this review, polyphenols can be absorbed in the stomach, the intestine, and colon, or be eliminated through feces.

### 2.1. Stomach

Studies performed in an in vitro model of the digestive tract showed the effects digestion exert on the metabolism and absorption of polyphenols. Once ingested, polyphenols can suffer mechanical disruption, which has been concluded to be a critical point in the liberation of the polyphenol: the chewing, the crushing in the stomach, the gastric movement, and friction between aliments facilitate the liberation and improve the absorption in the stomach [8]. Furthermore, it has been shown that an acidic pH does not affect the stability of the polyphenol and depending on the complex formed, aglycones could be liberated. On the other hand, mucins liberated during chewing and those present in mucus did not allow for the gastric absorption nor the liberation of the complex when the polyphenol presents an affinity toward them.

Some of the simple phenols and flavonoids can be found retained in the fiber of the extracellular matrix or united with polysaccharides on the cell wall by ester or hydrogenic bonds. These can be dissolved in alkaline environments such as in the small intestine, but not in the stomach. Some compounds are changed in the stomach such as procyanidin, which is degraded to epicatechin in the gastric juice, although this degradation is reduced in the presence of mucins by union or masking [8]. Additionally, flavonoids such as catechin and epicatechin are commonly found as aglycones and are easily liberated during gastric digestion as opposed to flavanols, which are almost exclusively found as β-glucosides, and are not liberated in the stomach. However, chlorogenic acids such as caffeic acid, ferulic acid, and p-coumaric acid are hydrolyzed in the stomach and thereby liberate aglycones [8], which permits their absorption. This has also been proven in in vitro studies that show how procinadin B1 and antocianines are absorbed at acidic pHs such as that of the stomach and are not absorbed at alkaline pHs such as that of the intestine [9].

The grand variety of polyphenols and the complexes they form complicate the study of their absorption on the stomach level. Nonetheless, generally, they can be absorbed in the stomach in their free forms but not when they form complexes. Even so, the major part of aglycones will be absorbed on an intestinal level.

### 2.2. Intestine

Studies have been performed on the metabolism and bioavailability of polyphenols in the small intestine in rats perfused with polyphenol representing plants including genistein, hesperetin, and ferulic acid [10].

The intestinal absorption of these flavonoids can be achieved through passive diffusion or co-transportation with sodium. The perfusion of various polyphenols did not affect the absorption, which suggests that the transport mechanisms cannot be saturated. Additionally, the intestinal cells have the capacity to generate phenolic complexes, which they can secrete to the lumen, thereby reducing the net absorption [10]. Previous studies have suggested that the active secretion of these complexes could be regulated by the MPR-2 and/or the P-glycoprotein. The MRP-2 is an ATP-dependent protein in charge of the detoxification of organic compounds bound to glutathione, glucuronidases, and sulfatases [11], and are mainly expressed in the liver or intestine.

It is important to keep in mind that the secretion rates of these complexes returning to the lumen are notably different depending on the polyphenol. For ferulic acid, it was low compared to the two flavonoids, which suggests a higher bioavailability for the latter. The secretion of these compounds combined can reach a saturation point, as the administration of higher quantities of polyphenols did not lead to a proportional increase in secretion. Other polyphenols such as genistein and hesperetin suffer hepatic conjugation and were secreted in bile once absorbed, which diminishes their bioavailability [10]. Furthermore, it has been observed how the intake of cranberry juice in humans shows different absorption patterns between individuals. Therefore, it must be taken into account that the absorption at the intestinal level is also determined by the patterns of the microbiome, which differs between each individual [12].

In conclusion, intestinal absorption, splanchnic metabolism, and biliary and intestinal secretion of polyphenols govern the majority of their bioavailability.

### 2.3. Colon

The polyphenols that are not absorbed in the stomach or small intestine can be metabolized by the intestinal microbiota and reach the colon where they can be absorbed or eliminated through feces.

The colon contains 10^12^ microorganisms/cm^3^ and therefore presents an immense catalytic and hydrolytic potential. Deconjugations happen easily, for example, quercetin-3-orham glucoside and quercetin-3-O-ramnoside cannot be hydrolyzed by human endogenous enzymes but are easily hydrolyzed in the microflora by organisms such as *Bacteroides distasonis* (α-ramnosidase and β-glucosidase), *B. uniformis* (β-glucosidase), and *B. ovatus* (β-glucosidase) [13].

As opposed to the enzymes in human tissue, the microflora of the colon can catalyze the decomposition of polyphenols to simple compounds such as phenolic acids. Other organisms such as enterococcus casseliflavus metabolize half of the sugar of quercetine-3-o-glucoside to make acetate and lactatebut does not metabolize aglycone. Studies in vitro have shown that when quercetin-3-O-ramnoside was incubated anaerobically with human intestinal bacteria, it was metabolized by 3,4-dihydroxilphenylacetic and 4-hydroxylbencoic acid. In humans, it has been observed how aglycone is transformed by the microflora of the colon to 3-hydrocylphenilacetic, 3-methoxi-4-hydroxyphenilacetic, 3,4-dihydroxylphenilacetic, 3,4-dihydroxitoluen, and b-m-hydroxyphenilhydrpacrilic acid [14].

A polymer of procyanidin also experienced degradation by the microflora due to the phenolic metabolic acids that are later absorbed in the colon [15]. The hydroxycinnamic acids connected to the cell walls of plans are not liberated by endogenous mammal enzymes and require enzymes such as xylanase and esterase from the colon microflora to be liberated [16].

Despite the great variety of polyphenols and metabolites they can form in the microflora of the colon, we have only cited a few examples in this revision. It is, however, clear that the polyphenols that cannot be absorbed or metabolized and reach the colon can, due to its microflora, experiment an active metabolism, thereby liberating aglycones and their derived metabolites, which can be further absorbed and enter the blood stream. Therefore, this pathway cannot be discarded when studying the pharmacological properties of polyphenols. Figure 2 summarizes the absorption and metabolism of polyphenols.

## 3. Bioavailability

Although we have already commented on bioavailability in the previous paragraphs, due to its complexity, we will proceed to summarize the known concepts. The chemical properties of polyphenols depend on their bioavailability, which can be measured indirectly by studying the antioxidant capacity in plasma or by directly measuring the quantity of polyphenol in plasma after the consumption of a compound in its pure form or foods containing the compound of interest. 

Studies in humans show that the intact quantities present in urine vary between 0.3 and 1.4% of phenolic compounds such as routine or glucoside of quercetin to 3–26% in catechins from green tea, soy isoflavones, citric flavanones, and anthocyanidins from red wine. However, the major part of the polyphenols ingested (75−99%) are not found in urine, implicating that they are not absorbed by the intestinal barrier nor metabolized by the microflora [3].

Studies have shown that intact flavonoid concentrations are rarely found to exceed 1 µM by a balanced diet. These concentrations are usually achieved 1–2 h post ingestion [17], except for the polyphenols that are absorbed only after their partial degradation by the colon microflora such as the routine liberated by quercetin, which reach peak plasmatic concentrations 7 h post ingestion [18]. Additionally, some compounds take up to one day for the body to eliminate such as equol, enterodiol, and enterolactone [19]. For example, after the consumption of flaxseed, plasma lignan concentrations did not peak until 9 h and were maintained for 24 h after intake, suggesting that the metabolism, absorption, and elimination of lignans from flaxseed takes longer than that of soy isoflavones.

As we have commented previously, the polyphenols present in food and drinks can be found naturally both in their free form and in a large variety of complexes. Additionally, they can be transformed all the way through the digestive tract from the mouth to the colon. They can be liberated from their conjunctions, re-conjugate in the intestine or liver, become metabolized by the microflora after being absorbed by the digestive tract, unite with plasmatic proteins, and are incorporated into adipose tissue. Due to the many possible structures and interactions, the bioavailability of polyphenols is extremely diverse. Understanding these processes can help us predict how, when, and what to ingest to obtain maximum efficiency from the beneficial and pharmacological properties, which we will describe in the following sections. 

### Experimental Studies

In the following paragraphs, we outline the most relevant studies performed in the last five years regarding polyphenols and their potential pharmacological effects (see Appendix A for search criteria and methodology).

## 4. In Vitro Studies

### 4.1. Cardiovascular Disease

Cardiovascular disease is the leading cause of morbidity and mortality, in 2015, the World Health Organization estimated that cardiovascular diseases accounted for over 30% of deaths worldwide [20].

It is commonly known that saturated free fatty acids lead to endothelial injury and cardiovascular complications [4]. A study by Song, J. et al. showed that 10, 50, and 100 µM resveratrol prevented palmitic acid-induced intracellular ROS by autophagy regulation through the AMPK-mTOR pathway, thereby providing cardiac protection in metabolic syndrome in human aortic endothelial cells [21]. Additionally, the resveratrol dimers ε-viniferin and pallidol at 10 and 50 µM exerted potential anti-angiogenic effects by enhancing eNOS activation via Akt phosphorylation and inhibition of the vascular endothelial growth factor receptor 2-induced PLCγ1 phosphorylation in human umbilical vein endothelial cells [22]. In another study on endothelial cells Xu, Y. et al. showed how tannic acid at 10 and 20 µM induces *KLF2* expression and attenuates TNFα-induced inflammation and monocyte cell adhesion [23]. 

The benefits of tea polyphenols have been widely studied both in vitro and in vivo. A recent study showed how pretreatment with1, 5, 10 µM epigallocatechin-3-gallate (EGCG), the major polyphenol found in green tea, improved cell survival after exposure to H_2_O_2_ in endothelial cells by inducing autophagy and inhibiting apoptosis by downregulating the PI3K-AKT-mTOR signaling pathway [24]. Additionally, 10, 30, and 50 µM of EGCG in neonatal rat cardiomyocytes have been shown to inhibit pressure overload-induced cardiac hypertrophy by upregulating the PSMB5/Nmnat2/SIRT6-dependent signaling pathway [25]. 

Moreover, medicinal plant extracts have long been used, especially in oriental cultures. Decoctions of 1 g/L teabag containing *S. cumini* and *P. mauritianum* have been shown to decrease ox-LDL uptake by more than 70%, while inhibiting the NF-κB pathway in the presence of pro-inflammatory concentrations of *E. coli* lipopolysaccharides in RAW 264.7 murine macrophages [26]. Furthermore, salvianolic acid A, a natural polyphenol extract widely used as traditional medicine in China for its cardioprotective functions, was recently shown to protect against lipotoxicity-induced myocardial damage at 5, 10, 20, and 40 µM through a TLR4/MAPKs signaling pathways in H9c2 cells [27].

### 4.2. Diabetes

There are few in vitro studies on the effects of polyphenols and their influence on diabetes type I. However, one study by Pham Hua et al. showed that a 4 µM tannic acid/poly(N-vinylpyrrolidone) multilayer coating attenuated the synthesis of pro-inflammatory cytokines and T cell effector responses involved in autoimmunity in vitro, thereby modulating the innate and adaptive immune response, and providing potential protective effects against type 1 diabetes [28]. 

Studies on type 2 diabetes, on the other hand, are more common. One study showed that polyphenolic extracts from Indian ginger at 6.25 µg/mL increased insulin stimulated glucose uptake in C2C12 cells by inhibiting α–amylase and α–glucosidase, decreasing glycation, and increasing *GLUT4* expression [29]. Similar results were found in a recent study of the effects of 30 and 90 μg/mL of the polyphenol-rich *Terminalia phaeocarpa,* which was found to inhibit pancreatic lipase, α-amylase, and α-glucosidase, TNF-α, IL-1β, and *CCL-2* expression in LPS-treated THP-1 cells [30]. A study by Vlavcheski et al. also showed how 5 µM rosmarinic acid increased glucose uptake to levels comparable to insulin and metformin treatments through AMPK phosphorylation in L6 rat muscle cells [31]. Furthermore, ellagic acid at 50 µM, 10 and 20 µM punicalagin, and particularly 50 µM of urolithin A have been shown to prevent metabolic diseases like obesity and diabetes by inhibiting lipase, α-glucosidase, and dipeptidyl peptidase-4 activity, and by reducing triglyceride accumulation and adipocyte formation through reducing adiponectin, PPARγ, GLUT4, and FABP4 expression in 3T3-L1 adipocytes [32].

### 4.3. Obesity

Obesity is rising worldwide and is mainly attributed to changes in lifestyle including overconsumption of food and decreased physical activity. Studies of plant extracts and compounds show that polyphenols can play a beneficial role in the prevention of obesity by reducing cholesterol levels, fatty acid synthesis, triglyceride formation, and inhibiting genes involved in adipocyte differentiation and triglyceride accumulation [33].

Several studies have shown that ginger enhances thermogenesis, energy expenditure, and provides lipid metabolism regulation, thereby providing potential therapeutic effects on obesity. The polyphenol 6-gingerol improves mitochondrial respiration and energy metabolism by increasing the uncoupled oxygen consumption rate of protons leaked and upregulating mitochondrial biogenesis markers through the AMPK signaling pathway in 3T3-L1 and HepG2 cells at 47.81 ± 0.62 mg/g and 10 to 200 µM quantities, respectively [34,35]. Similar results were found when analyzing the effects of ginseng, which was also found to exert anti-obesity effects by inducing adipocyte browning through AMPK activation at 25, 50, and 100 μg/mL in 3T3-L1 and primary white adipocytes [36]. Additionally, anti-obesity properties of phenolic compounds of canola have been studied using adipogenic differentiation inhibition of a murine mesenchymal stem cell line (C3H10T1/2). Significant adipogenesis and pancreatic lipase inhibitory activities were found after treatment of 2–3 mg/mL canola meal extract through the downregulation of the peroxisome proliferator-activated receptor gamma (PPARγ) signaling pathway [37].

In 2019, Mao et al. found that the 10 and 20 µM EGCG exerted suppressive effects on lipid accumulation and proinflammatory cytokines by inhibiting the JAK2/STAT3 signaling pathway in palmitic acid-stimulated BV-2 microglia [38]. Furthermore, 2 and 10 µM curcumin have been found to repress adipogenic differentiation through the Wnt signaling pathway in 3T3-L1 cells [39]. Finally, 50 µM of piceatannol, a lesser studied analogue of resveratrol, has been proven to have antiadipogenic properties by limiting lipid synthesis and accumulation in hMSC-derived adipocytes, lowering glucose transport into adipocytes and inhibiting the PPARγ pathway, and thereby can be used in the treatment of metabolic complications associated with obesity [40].

### 4.4. Digestive Diseases 

The intestinal epithelium is in direct contact with a variety of nutrients, microbes, and exogenous toxins, offering both a barrier and absorptive functions in the intestine [41]. Punicalagin, a major polyphenol abundant in pomegranate fruit, has been found to increase cell viability at 40 µM in small intestine epithelial rat IEC-6 cells in response to heat shock via the PI3K/Akt pathway, thereby preserving the integrity of the intestinal epithelium [41].

Lipid metabolism disorders in liver often lead to nonalcoholic fatty liver diseases (NAFLD), which is a common chronic liver disease connected with the accumulated uptake of free fatty acids. A study found that 200 μM cichoric acid limits lipid accumulation in HepG2 cells by enhancing the Akt/GSK3β signaling pathway and modulating the downstream expressions related to lipid metabolism, indicating that cichoric acid can be useful as a natural NAFLD modulator [42].

The beneficial health effects of walnut have long been attributed to its rich *n*-3 polyunsaturated fatty acid character and anti-oxidative and anti-inflammatory properties. In a study by Park et al., 10 or 20 µg/mL walnut polyphenol extracts were found to suppress *H. pylori* infection induced phosphorylation of STAT3 in a PPAR-γ dependent manner in RGM-1 gastric mucosal cells, thereby providing protection against gastric pathologies such as chronic atrophic gastritis and gastric cancer [43]. Moreover, Hossen et al. found that 50 µM resveratrol extract exerted anti-inflammatory effects on lipopolysaccharide-treated macrophages by inhibiting the signaling events upstream of NF-κB translocation such as phosphorylation of AKT and the formation of PDK1-AKT signaling complexes [44].

To this day, the only treatment for celiac disease is a life-long, strict gluten-free diet, however, peptide antibodies have become gradually more important in the diagnostic work-up of celiac disease [45]. In one study, 80 μM of procyanidins were shown to reduce the celiac disease bioactive peptide 32-mer translocation in Caco-2 cells, thus modulating the transepithelial transport of celiac disease bioactive peptides [46].

### 4.5. Neurodegenerative Disease

The health effects of the consumption of polyphenols through a diet rich in fruits and vegetables and their impact on neurodegenerative diseases have been extensively studied. *Ribes diacanthum* Pall, a native Mongolian medicinal plant, has been reported to show antioxidant activities due to its polyphenolic content, exert cytoprotective effects at 10, 25, and 50 μg/mL in mouse hippocampal neuronal HT-22 cells by regulating the expression of antioxidant enzymes and activating the BDNF/TrkB pathway [47]. In another study where PC12 cells were induced to apoptosis, 10 μM of Honokiol, a magnolia derived polyphenol, was shown to promote neuroprotective effects by increasing glutathione levels, upregulating cytoprotective proteins, and promoting transcription factor Nrf2 nuclear translocation and activation [48].

One study on the bioavailability of 19 plant metabolites and their effects on oxidative stress related neurodegeneration found that 10 μM 3,4-dihydroxyphenylpropionic acid, 3,4-dihydroxyphenylacetic acid, gallic acid, ellagic acid, and urolithins decreased oxidative stress-induced apoptosis by preventing caspase-3 activation through the mitochondrial apoptotic pathway in SH-SY5Y cells [49].

Neurodegenerative diseases are typically characterized by the loss of neurons due to several pathological mechanisms or processes such as glutamate excitotoxicity, oxidative stress, and abnormal apoptosis [50]. Resveratrol is well known for its antioxidative properties, however, a study performed by Wang, W. et al. demonstrated that 50 mg/mL resveratrol can exert additional protective effects against neurodegeneration by inhibiting glutamate-induced apoptosis in primary hippocampal neurons by enhancing Bcl-2 and decreasing Bax and Caspase-3 expression through the PI3K/AKT/mTOR pathway [51]. Furthermore, 100 μM resveratrol has been found to exert neuroprotective effects by combatting glutathione depletion through the HO-1 pathway in C6 astroglial cells [52].

### 4.6. Cancer

Many interesting findings have also been published regarding the protective benefits of polyphenols against cancer. Studies have shown how treatment with 140 µM tea polyphenols induces mitochondrial-mediated apoptosis by activating the caspase-3 and PARP apoptotic cascade in human gallbladder cancer cells [53], 0.5–5 μg/mL of EGCG reduces the cell viability and increases the apoptosis rate of myeloid-derived suppressor cells, mainly through the Arg-1/iNOS/Nox2/NF-κB/STAT3 signaling pathway [54], and 15 μM EGCG inhibits Excision Repair Cross-Complementation Group1/Xeroderma Pigmentosum group F, thereby inhibiting DNA repair and enhancing cisplatin sensitivity in human lung cancer cells [55]. A total of 0.1–1 mM catechin with lysine has also been proven to have antimigratory effects in breast, pancreatic, and colorectal cell lines through JAK2/STAT3 and Wnt pathway inhibition [56]. Additionally, 500 µM of Annurca apple polyphenol extracted catechin has been shown to be a potent pro-oxidant and antiproliferative agent able to downregulate the ERK1/2 pathway, leading to cell cycle inhibition and apoptosis [57]. The flavonoid quercetin, a polyphenol abundant in the Mediterranean diet, also has apoptotic effects through the regulation of the phosphatidylinositol 3-kinase (PI3K)/protein kinase B (AKT)/forkhead box protein O 1 (FOXO1) pathway in A549 cells at 160 μg/mL [58]. Furthermore, Ruzzolini, J. et al. showed that oleuropein, a glycoside found in olive leaves at 250 µM affected cell proliferation and induced the downregulation of the pAKT/pS6 pathway in Human BRAF Melanoma Cells [59].

As seen, polyphenols have the ability to exert several beneficial effects in vitro. Figure 3 summarizes the effects discussed in this review.

## 5. Animal Models

### 5.1. Cardiovascular Disease 

Cardiovascular diseases are arguably the most important comorbidities in chronic obstructive pulmonary disease [60]. Curcumin, a dietary polyphenol found in turmeric, has been found to have therapeutic benefits in chronic obstructive pulmonary disease by inhibiting NF-κB. In a recent study, administration of 2.5, 5, and 7.5 μM and 100 mg/kg^−1^ curcumin attenuated cigarette smoke-induced inflammation both in vitro and in vivo by modulating the PPARγ-NF-κB pathway [61]. Furthermore, curcumin at concentrations of 25 mg/kg/day has been found to reduce serum lipid levels, inflammation damage in heart, lungs, and kidneys, and decrease the formation of atherosclerotic plaque in the aorta through HMGB1-TLRS-NF-κB signaling pathway-related proteins in a high fat diet fed mice [62]. Protocatechuic acid has also been reported for its cardiovascular-protective effects. A study showed how 200 mg/kg/day for 12 weeks significantly improved endothelium-dependent IGF-1-induced vasorelaxation by enhancing the activity PI3K/NOS/NO pathway in aged hypertensive rats administered [63].

Additionally, mangiferin, a polyphenol found in mango and papaya, has been found to decrease plasma lipids and inflammatory levels in mice with high fat diet-induced vascular injury through the PTEN/Akt/eNOS signaling pathway at 20 µM [64]. Moreover, in an in vivo myocardial ischemia rat model, 200 mg/kg lychee extract was found to exert myocardial protection through bcl-2/bax gene expression [65].

### 5.2. Diabetes Mellitus

Numerous studies have been performed to analyze the effects of polyphenols on diabetes. Regarding type 1 diabetes mellitus, a study was performed in streptozotocin-induced diabetic mice to evaluate the effects of 10–250 mg/kg (-)-epicatechin-3-O-β-D-allopyranoside from Davallia formosana (BB) over a 28-day period. BB was found to increase the expression of fatty acid oxidation enzymes such as PPARα and carnitine palmitoyl transferase 1a and lower mRNA levels of SREBP1c and AFABP2, thereby reducing plasma triglyceride levels [66]. Caesalpinia bonduc, traditionally used in herbal medicines for the treatment of a wide range of diseases, has been shown to restore insulin, glucose, leptin, and amylin as well as carbohydrate metabolizing enzymes after eight weeks of 250 and 500 mg/kg administration in alloxanized diabetic rats by inhibiting the JNK signaling pathway [67].

To study the effects of gallic acid on type 2 diabetes, a combined mouse model of a high fat diet induced obesity and low dose streptozotocin induced hyperglycemia was used as it mimics the pathological condition. The administration of 150 mg/kg gallic acid was shown to alleviate lipid accumulation through the upregulation of *β*-oxidation and ketogenesis [68]. Another study showed that the administration of 50 mg/kg ellagic acid for 45 days mitigated insulin resistance in T2DM rats by reducing HIF-α and upregulation of NRF2 by enhancing the Akt signaling pathway [69]. Muscle loss is highly associated with diabetes as elevated lipid metabolites impair myogenesis. Dietary supplementation of 20 and 200 mg/kg oligonol for 10 weeks has been found to decrease lipid levels and attenuate muscle loss by downregulating the MAFbx/*atrogin*-*1* pathway and upregulation the SIRT1/AMPKα pathway in diabetic *db/db* mice [70].

### 5.3. Obesity 

In an in vivo study in old mice, 2.3 μg/kg/day resveratrol supplementation for two days was shown to activate fatty acid degradation and inhibit their synthesis through activation of the AMPK pathway. The respiratory quotient was also found to be lowered due to fatty acid transport from white adipose tissue into the mitochondria and subsequent β-oxidation in muscles and liver, determined by increased expression and phosphorylation of the enzymes involved in lipid catabolism such as acyl-CoA synthetase, carnitine acylcarnitine translocase, and carnitine palmitoyl transferase 1 [71]. Similarly, another study reported that 1000 mg of chokeberry polyphenols extract/kg improved dyslipidemia and lipid metabolism and lipogenesis mediated through the FXR and TGR-5 signaling pathway on high fat diet-fed rats [72]. Furthermore, 100 mg/kg cinnamon polyphenol extracts have been found to decrease body weight and visceral fat as well as improve lipid profile and MDA concentrations via the PPAR-α and Nrf2 pathways in rats fed a high fat diet [73]. The administration of curcumin also reduces body weight and fat accumulation, lowers blood insulin levels, and preserves islet integrity through the downregulation of TXNIP transcription factors, as reported in a study on mice administered a high fat high sugar diet supplemented with 0.4% (*w/w*) curcumin [74]. Additionally, the interaction between prebiotics and probiotics might exert synergistic benefits on health. Cho et al. investigated the combined effects of 10% polyphenol-rich wine-grape seed flower and lactic acid bacteria from kefir on high-fat induced obese mice. They reported that this combination exerted the highest improvements on body weight gain, plasma insulin, and cholesterol concentrations compared to their administration separately and the control [75].

### 5.4. Neurodegenerative Diseases

Coffee polyphenols such as chlorogenic acid halts neurodegeneration and provides neuroprotection against Parkinson’s disease by blocking the dopaminergic neuronal toxin 6-OHDA and combinatorial α-synucleinopathy through the Akt1-CREB-RNF146 pathway at 1 and 10 µM [76]. Moreover, 5-caffeoylquinic acid, another polyphenol present in coffee, exerts neuroprotective effects by reducing Aβ plaque formation and neuronal loss and ameliorating cognitive decline by modulating the Aβ clearance pathways when administered with a high fat diet at 0.8% (*w*/*w*) compared to the control [77]. Similarly, 0.5% rosmarinic acid has been found to suppress Aβ accumulation and increase monoamines such as norepinephrine, 3,4-dihydroxyphenylacetic acid, dopamine, and levodopa in mice by enhancing the dopamine signaling pathway [78]. Resveratrol has also been found to exhibit neuroprotective effects against Parkinson’s disease. However, due to its water solubility, its oral bioavailability reduces and therefore demonstrates low efficacy in the blood and brain. In a recent study, a nanocrystal formulation of resveratrol was developed to enhance its oral bioavailability and delivery to the brain. They reported higher plasma and brain concentrations, attenuated dopamine deficiency, and notable improvements in behavior. Immunoblot analysis revealed that these benefits are mediated by the Akt/Gsk3β signaling pathway [79].

Although spinal cord ischemic stroke is not considered a neurodegenerative disease, it is known to produce detrimental effects on neuronal function, leading to the development of neurodegenerative diseases. One of the important pathophysiological effects of ischemic stroke is apoptosis. Intraperitoneal injection of resveratrol at 30 mg/kg has been found to provide neuroprotective effects against cerebral ischemic injury by attenuating neuronal apoptosis by upregulation of the PI3K/Akt/mTOR pathway through the activation of the JAK2/STAT3 pathway [80]. Additionally, rats treated with 10 mg/kg resveratrol showed improvements in injury percussions after spinal cord ischemia-reperfusion injury by its ability to suppress the activation of the iNOS/p38MAPK pathway [81]. Ellagic acid has also been shown to protect against ischemia by increasing the proliferation of NSCs through the Wnt/β-catenin signaling pathway, thereby providing protection against nerve dysfunction at concentrations of 1.0, 3.0, and 9.0 μg/mL [82].

### 5.5. Digestive Diseases 

NAFLD is associated with hepatic insulin resistance (IR). Sub-chronic pretreatment of 20 mg/kg resveratrol in Winstar rats fed a high fat diet has been found to prevent early hepatic IR through inhibiting PKC/JNK activation, thereby reducing the expression of acyl-CoA:glycerol-sn-3-phosphate acyltransferase and diacylglycerol:acyl-CoA acyltransferase, two critical enzymes in the glycerol-3-phosphate pathway for de novo triglycerides synthesis [83]. Autophagy induced by the SIRT1-FOXO signaling pathway has recently been shown to be a critical protective mechanism against NAFLD development. Ding et al. found that 200 mg/kg bw resveratrol combined with a caloric restriction of 30% increased the expression of SIRT1 and autophagy markers while decreasing ER stress markers in the liver, partially preventing hepatic steatosis and hepatocyte ballooning, and alleviating lipid metabolism disorder in rats fed a high fat diet [84]. Similarly, 100 mg/kg curcumin attenuates hepatic steatosis and reversed the abnormalities of serum biochemical parameters of non-alcoholic fatty liver disease induced by a high-fat and high-fructose diet in C57BL/6 mice via the Nrf2-FXR-LXR pathway [85].

*Moringa oleifera* Lam. is an essential herb used in the treatment of various diseases such as inflammation, diabetes, and high blood pressure. Quantities of 50 mg/kg and 200 mg/kg of *M. oleifera* polyphenol extract, rich in quercetin, kaempferitrin, chlorogenic acid, luteolin, and rutin, has been found to alleviate the symptoms of dextran sulfate sodium-induced colitis by the downregulation of NF-*κ*B *p65* and p-I*κ*B*α* [86].

### 5.6. Cancer

Red wine consists of a large number of compounds including resveratrol, which exhibits chemopreventative and therapeutic effects. However, a recent study reported that another polyphenolic red wine component, ellagic acid, inhibited lung cancer cell proliferation at an efficacy approximately equal to that of resveratrol at concentrations higher than 25 µM by inactivation of the mechanistic target of rapamycin signaling pathway and significant inhibition of tumor growth with the suppression of CIP2A levels and increased autophagy in tumor-bearing mice [87]. A quantity of 20 µM of ellagic acid has also been found to suppress lung metastasis and exerts beneficial effects against endometrial cancer in BALB/c nude mice via the PI3K signaling pathway [88]. 

As mentioned earlier, the tea polyphenol EGCG is widely known for its anti-inflammatory effects. In a recent study, 80 µM EGCG was shown to inhibit lung cancer cell proliferation by suppressing NF-kB signaling in a mouse xenograft model, which was potentiated with coadministration by the NF-κB inhibitor BAY11-7082. The scientists of this study concluded that this synergistic effect of low dose EGCG at 2.5 or 5 µM with BAY11-7082 may serve as a novel therapeutic strategy for lung cancer [89]. In another study, mango polyphenols (0.8 mg gallic acid equivalents per day) and pyrogallol (0.2 mg/day) administered for four weeks to mice xenografted with MCF10DCIS.com cells inhibited proliferation of breast cancer cells through ROS-dependent upregulation of AMPK and downregulation of the AKT/mTOR pathway [90]. Phenolic extract from olive trees also decreased HCT116 and HCT8 colon cancer cell viability by inducing the mitochondrial apoptotic pathway when treated with 20 μg/mL [91]. Finally, in another study on a mouse xenograft model, curcumin treatment was shown to inhibit hepatocellular carcinoma proliferation in vivo at concentrations of 50 and 100 mg/kg by decreasing VEGF expression and PI3K/AKT signaling in a mouse xenograft model [92].

Similar to the findings in in vitro studies, polyphenols have also been found to exert therapeutic effects in animal models. The effects outlined in this review are summarized in Figure 3.

## 6. Human Studies

### 6.1. Cardiovascular Diseases

In clinical trials aimed at attributing the therapeutic effect of polyphenols in cardiovascular diseases, the changes in arterial blood pressure (BP), lipid metabolism, endothelial function, and other risk factors were tested as the endpoint results of various interventions.

Significant reduction of diastolic BP and the decrease in systolic BP were confirmed in participants with stage 1 or 2 hypertension and metabolic syndrome after 28-day treatment with supplements containing several phenolic compounds (grape seed and skin 330 mg, green tea extracts 100 mg, quercetin dehydrate 50 mg, resveratrol 60 mg, and other phytochemicals 10 mg a day). The complementary analysis allowed researchers to assume that the observed effect on BP could be related to an increase in nitric oxide (NO) bioavailability [93]. Results in a crossover trial in Finland showed a statistically significant diastolic BP lowering effect and decrease in IL-10 and TNF-α after the eight week consumption of Aronia mitschurinii products with the total mean intake of 2194 mg/day chokeberry polyphenols [94]. A randomized, placebo-controlled study involving participants with stage 1 hypertension tested the potential hypotensive effect of low molecular weight oligomeric procyanidin extract (150 mg/day) made of French Maritime Pine bark. When BP changes were compared to those of the placebo group, significant lowering of systolic BP in the result of 5-week treatment was confirmed only in female participants. The supplementation of procyanidins was also associated with beneficial changes in lipid profile of a statistically significant reduction in the Apo B-100/Apo A-1 ratio and increase in high-density lipoprotein-cholesterol [95]. Additionally, favorable effects on BP and plasma lipids resulting from phenolic-rich olive leaf extract intake (136.2 mg oleuropein and 6.4 mg hydroxytyrosol daily) was confirmed in a randomized trial. After a 6-week treatment, both systolic and diastolic BP, total cholesterol (TC), low-density lipoprotein cholesterol (LDL-C), and triglycerides were statistically significantly lowered in pre-hypertensive participants when compared with values in the control group [96]. To analyze whether a combined polyphenol preparation could attenuate cardiovascular disease risks in healthy adults, citrus, bitter orange, grapefruit, and olive polyphenol compounds in a daily dose of 250–300 mg flavanone-glycosides, 175–200 mg of flavones, and 85–90 mg secoiriodids and phenolics were administered. After eight-weeks of treatment, a decrease in mean values of diastolic BP, TC, LDL-C, LDL-oxidase, IL-6, and protein carbonyl as well as increased flow-mediated vasodilation and reduced/oxidized glutathione ratio in the active product group were statistically significant compared to the placebo group [97]. A study on the intake of a polyphenol-rich (around 210 mg/day) cranberry beverage by overweight or obese adults with CVD risk led to statistically significant changes in lipid metabolism, pro-inflammatory markers, and modulators of vascular tone in the intervention group compared to the control. After eight-weeks, increased HDL-H and lowered CRP were shown, whereas the beneficial effects of augmented reduced/oxidized glutathione ratio, plasma endothelin-1, and NO levels were confirmed only at the baseline week [98]. In patients with peripheral arterial disease of lower limbs and intermittent claudication, a 24-week treatment with 2 g/day of the polyphenolic extract of Annurca apple significantly increased walking autonomy and improved hemodynamic parameters such as an ankle-brachial index and acceleration time to peak systolic velocity. At the same time, no significant differences were revealed in the placebo group. Additionally, self-reported symptoms like paresthesia, cold limbs, and cold toes caused by vascular abnormalities in the treatment group diminished or disappeared, whereas no similar changes were observed in the placebo group [99]. Figure 4 summarizes the effects of polyphenols on cardiovascular disease outlined in this review. 

### 6.2. Diabetes Mellitus

The potential of natural polyphenols to ameliorate glucose metabolism and decrease complications of diabetes mellitus were evaluated in various clinical studies involving patients with type 2 diabetes mellitus (T2DM) or non-diabetic adults with risk factors for T2DM such as overweight, prediabetes, low physical activity, and family history. Usually, the parameters of inflammation, glucose and lipid metabolism, and antioxidant status in participants are measured as indicators of the effect of polyphenol treatment. In a double-blind, randomized, placebo-controlled trial, the intake of 40 or 500 mg/day of resveratrol was tested for anti-inflammatory effects and improvement in total antioxidant status (TAS) in T2DM patients. After six months of treatment, the C-reactive protein (CRP) concentration decreased more in the high dose resveratrol group than in the low dose group, although none of the results were statistically significant compared to those of the placebo group. A more prominent decrease in CRP from the baseline was associated with lower diabetes duration and some additional modifying factors [100]. The concentration of an acute-phase protein, pentraxin, whose role in inflammatory responses remains unclear, was increased after intervention, meanwhile significant improvement of TAS was observed after supplementation with resveratrol [101]. To prove the anti-inflammatory and lipid regulatory activity of curcumin, which could be used to prevent complications of diabetes, a study tested inflammatory markers, lipid profile, and serum adiponectin changes in T2DM patients after treatment. Results confirmed that the intake of 1500 mg/day curcumin for ten weeks significantly decreased high-sensitivity CRP and triglyceride levels, and increased anti-inflammatory cytokines and serum adiponectin levels compared to the placebo [102]. The beneficial effect of apple polyphenols on glucose homeostasis was observed in a trial conducted in Japan. In non-diabetic adults with high-normal and borderline plasma glucose level, a 12-week administration of various apple polyphenols at 600 mg/day improved the results of the oral glucose tolerance test (OGTT), showing statistically significantly lower results in measurement after 30 min and the AUC of plasma glucose concentration in OGTT than in the placebo group subjects [103]. According to the results of a trial involving overweight adults with confirmed insulin resistance, a 6-week supplementation of strawberry and cranberry polyphenol extracts of 333 mg/day had a favorable impact on insulin sensitivity. In said study, the whole-body insulin sensitivity was determined by the hyperinsulinemic–euglycemic clamp method, and statistically significant improvement in insulin sensitivity was observed in the polyphenol treatment group, although no beneficial changes in lipid profile, oxidative, and inflammatory markers were confirmed [104]. Figure 5 below summarizes the effects of polyphenols on diabetes outlined in this review.

### 6.3. Obesity

Influence on body composition parameters, blood lipids, and hormones regulating energy intake are most common control parameters in clinical trials aimed at investigating the anti-adiposity potential of polyphenols. 

Body mass index (BMI), body fat percent, and abdominal circumference decreased in overweight and obese adults over two months following an isocaloric diet and consuming a daily dose of the extract of *Hibiscus sabdarifa* L. and *Lippia citriodora* L. containing 500 mg phenolic compounds. Improvement in these anthropometric measurements was more prominent in overweight than obese subjects and also significantly differed from the results in the placebo group [105]. This same phenolic preparation was studied regarding its effect on appetite regulation. The results revealed a decrease in the hunger-stimulating hormone ghrelin, self-evaluated hunger sensation, and an increase in glucagon-like peptide-1 in the treatment group [106]. In a trial conducted in Korea, anthropometric indicators were evaluated after a 12-week intake of 900 mg/day citrus flavonoids containing preparation Sinestrol-XPur. The overweight and obese adults in the intervention group had a statistically significant decrease in BMI and body fat mass compared to the controls [107]. A study on healthy adults observed that a 10-day intake of a beverage containing 165 mg black tea polyphenols statistically significantly increased fecal lipid excretion in the treatment versus placebo group, but did not cause any changes in the anthropometric parameters of the study participants [108]. A double-blind, randomized trial confirmed beneficial effects of bergamot juice flavonoid preparation of 650 mg or 1300 mg per day in the study group of patients with metabolic syndrome. The 90-day long treatment was associated with significant body weight and BMI reduction and beneficial changes in several parameters of blood lipids such as decreased TC, TG, and atherogenic index of plasma [109]. Figure 6 below summarizes the outlined effects of polyphenols on obesity.

### 6.4. Neurodegenerative Diseases

Antioxidant activity, anti-inflammatory, and neuroprotective properties of polyphenols allow us to consider them as promising candidates in preventing and treating neurodegenerative diseases. Nevertheless, evidence is scarce on the neuroprotective or therapeutic effects in human intervention trials.

Results in a comprehensive neurocognitive test did not show significant benefits from the supplementation of a walnut-enriched diet (accounting for 15% of the daily energy intake) on cognitive function in a large cohort of cognitively healthy seniors. Although walnuts were chosen as the nuts richest in polyphenol, the two-year intervention did not significantly improve global cognitive composite scores nor of specific cognitive domains such as language or memory. However, higher effectiveness of the intervention was observed in a subgroup of participants with lower education levels and lower background status of dietary α-linolenic acid intake [110]. A pilot study reported results of a one-year intake of 10 mg/day resveratrol in participants with mild to moderate Alzheimer’s disease. Less deterioration of scores according to Alzheimer’s disease assessment scales was found in the intervention group than in the control group, but none of the results were statistically significant [111]. A multicenter trial conducted in Germany tested whether the intake of epigallocatechin in an increasing dose of 400 to 1200 mg/d could modify disease progression in multiple system atrophy patients. Results did not confirm a statistically significant change in the motoric examination score of the Unified Multiple System Atrophy Rating Scale after 48-week treatment compared with the placebo. A MRI sub-study of disease-affected brain regions, however, indicated substantial lower annual striatal volume loss in the intervention than in the placebo group [112]. Figure 7 below summarizes the outlined effects of polyphenols on neurodegenerative diseases in humans.

### 6.5. Digestive Disease

Several studies have reported the beneficial effects of curcumin supplementation on patients with hepatic steatosis and NAFLD. In a randomized double-blind placebo-controlled trial, patients with NAFLD were randomly assigned to receive an amorphous dispersion curcumin formulation of 70-mg curcumin/day or placebo for a period of eight weeks. Compared with the placebo, curcumin was associated with a significant reduction in liver fat content (78.9% improvement in the curcumin vs 27.5% improvement in the placebo group) as well as significant reductions in BMI and serum levels of TC, LDL-C, triglycerides, aspartate aminotransferase, alanine aminotransferase, glucose, and glycated hemoglobin [113]. In another study, fifty-eight NAFLD patients participated in a randomized, double-blind, placebo-controlled study. The subjects were allocated randomly into two groups, receiving either a capsule of 250 mg phospholipid curcumin or placebo for a period of eight weeks. Compared with the placebo, supplementation resulted in significant decreases in serum levels of 3-methyl-2-oxovaleric acid, 3-hydroxyisobutyrate, kynurenine, succinate, citrate, α-ketoglutarate, methylamine, trimethylamine, hippurate, indoxyl sulfate, chenodeoxycholic acid, taurocholic acid, and lithocholic acid [114]. Furthermore, in another trial, where a total number of 80 patients were randomized to receive either 250 mg curcumin daily or the placebo for two months, the grade of hepatic steatosis, and serum aspartate aminotransferase levels were significantly reduced in the treatment group compared to the placebo [115].

Additionally, in a double-blind placebo controlled clinical trial of 102 patients with liver steatosis, the intervention group receiving 300 mg/day of a nutraceutical containing a bergamot polyphenol fraction and artichoke extract for 12 weeks presented a greater reduction in liver fat content than the placebo group. The percentage CAP score reduction was also statistically significant in those with android obesity, overweight/obesity as well as in women. However, after adjustment for weight change, the percentage CAP score reduction was statistically significant only in those over 50 years old [116]. 

Pediatric Crohn’s disease (CD) has been linked with higher thromboxane levels. On this note, in a small-scale study with 14 children suffering from CD and 15 healthy controls, the patients received polyphenolic extract pycnogenol to investigate the relationship between polyphenols, thromboxane levels, and oxidative stress in children with Crohn´s disease. At baseline, CD patients had significantly higher levels of the static and dynamic forms of thromboxane B2 in comparison to the controls. After 10 weeks, treatment with pycnogenol decreased the level of the dynamic form of thromboxane B2, indicating that pycnogenol administration may positively influence clinical symptoms of CD such as thromboembolic episodes [117]. In a large-scale study of 110 patients with CD (73% women) and 244 patients with ulcerative colitis (57% women), 440 and 976 controls aged 20 to 80 years from eight countries were recruited between 1991 and 1998. During the follow-up period of up to December 2010, the intake of flavones and resveratrol reduced the risk of CD, although no significant associations were found for dietary polyphenol intake and ulcerative colitis [118]. Figure 8 summarizes the effects of polyphenols on digestive diseases seen in the outlined human studies. 

### 6.6. Cancer

Several human experimental studies have been performed with the aim to prove the potential of polyphenols as a preventive tool as part of combined treatment therapy to attenuate the anticancer effect of conventional chemotherapeutics and as drugs to reduce the side effects caused by chemotherapy. Polyphenols have been widely studied, and anticancer effects are well explained in animal models and cell culture experiments. The molecular mechanism of action of the experimental cell lines derived from specific cancer cells could prove to be similar to the mechanisms in the human body as a whole. However, the interpolation of results to humans should be done with caution due to the absence of endogenous metabolism in cell models and the differences in gut microflora interaction with polyphenols in animal studies [119].

#### 6.6.1. Colorectal 

The trial conducted by Sincrope et al. in a group of patients with colorectal adenoma or cancer diagnosis did not prove the effect of 1200 mg daily green tea polyphenol preparation containing 65% EGCG and 25% other catechins. There was no significant decrease in adenoma recurrence rate and no reduction in percent of aberrant crypt foci, defined as precursors of colorectal cancer in the study group comparing to the placebo group after 6-month treatment [120].

A multicenter study in Germany examined the effect of 3-year treatment of 300 mg per day EGCG on colorectal adenoma incidence rate in patients after colorectal polypectomy. The treatment group showed a slightly lower incidence rate than in the placebo group though the difference was not statistically significant [121]. Intending to test whether the effects of ellagitannins to modify colon gene expression observed in vitro could be reproduced in humans, the trial involved patients with malignant colon biopsy with 5–35 days supplementation with ellagitannins and ellagic acid-containing pomegranate extract capsules. Analysis of particular colorectal cancer-related gene expression in malignant and normal colon tissue after pomegranate extract treatment did not show changes concordant to those observed in preclinical studies [122].

#### 6.6.2. Prostate

A randomized controlled trial of prostate cancer prevention assessed the green tea catechin metabolite plasma level and evaluated the changes in prostate-specific antigen (PSA) in men with an increased risk of prostate cancer (PSA in the range between 2.0–2.974 ng/mL or 2.974–19.95 ng/mL and negative biopsy). The green tea polyphenol intake was measured either as intake by drinking at least three cups of green tea daily, or by capsules containing 600 mg flavan-3-ol EGCG daily. At six months, EGCG plasma concentration was significantly higher than baseline in both intervention groups, while in the placebo group, changes were not detected. The PSA level, reported as one of the secondary endpoints, did not differ between the active peroral intake and placebo groups nor in each group compared to the baseline [123].

The therapeutic potential of muscadine grape skin supplement in prostate cancer patients with biochemically recurrent (BRC) prostate cancer was studied in a randomized, multicenter, placebo-controlled trial. Participants in both intervention groups received powdered muscadine grape skin with the main phenolic component ellagic acid as preparation MuscadinePlus in doses of 500 mg or 4000 mg daily for six to 12 months. The response to the therapy was measured by changes in clinical markers—PSA level in blood plasma and PSA doubling time (PSADT) with respect to individual PSADT calculated from PSA results in a one year time period prior to enrollment in the trial. Decrease in PSA ≥50% of baseline from trial and prolongation of PSADT resulting from therapy was defined as a clinically significant effect of grape skin phenols as a treatment and would allow delaying androgen deprivation hormonal therapy in men with BRC prostate cancer. Results did not confirm statistically significant differences in the course of disease or increase in PSADT in any group of MuscadinePlus doses compared with the placebo. Additional genetic analysis of single nucleotide polymorphism (rs4880) of superoxide dismutase-2 gene revealed a more significant increase in PSADT in high-dose MuscadinePlus patients with SOD2 Ala/Ala genotype than patients in the control group [124].

#### 6.6.3. Bladder

A randomized, double-blind, placebo-controlled study enrolled non-invasive urinary bladder cancer patients before surgical treatment and studied the effects of the intake of polyphenon E containing 800 mg or 1200 mg EGCG. Accumulation of EGCG in excised bladder tissue was determined in a dose-response relationship in normal and malignant tissue and could suggest that EGCG induces local biological activity in bladder tissue. Analysis revealed statistically significant downregulation of two specific tissue biomarkers of cellular proliferation–clusterin and proliferating cell nuclear antigen in both groups of study participants after the 14 to 28 days course of polyphenon E intake. This pilot study shows that EGCG has the potential for use as an anticarcinogenic agent in bladder cancer [125].

#### 6.6.4. Side Effects of Anticancer Therapy 

The potential of topically administered silymarin flavonolignans to prevent the development of radiodermatitis as a complication of radiotherapy was tested in a trial with breast cancer patients. Study results showed that a three week application of silymarin gel on the skin of the chest wall delayed the development of radiodermatitis and decreased its severity compared to results in the placebo group [126]. These beneficial effects of silymarin were also observed in patients with gastrointestinal cancer. After nine weeks of application of silymarin gel, the frequency of severe hand-foot syndrome cases caused by the chemotherapy drug capecitabine was statistically lower in the intervention group compared to the placebo [127]. Figure 9 summarizes the effects of polyphenols on cancer seen in the outlined human studies.

## 7. Conclusions

Based on the data collected in this review, we can conclude that various plants in our diet contain phenolic compounds that have pharmacological activities against diseases such as cardiovascular, digestive, and neurodegenerative disease, diabetes, obesity, and cancer. In vitro studies showed clear and specific mechanisms of action when the polyphenol was used in its pure form and at high concentrations compared to when ingested through foods rich in these polyphenols. The same was observed when treatments were carried out on experimental animals. In humans, the mechanisms of action are more complicated to obtain, although favorable biochemical and physiological parameters were observed against some of the pathologies studied when pure polyphenol extracts or aliments rich in these polyphenols are administered. These effects in vitro, in animals and humans are all outlined in Figure 10 below. Nonetheless, the bioavailability that they present when we eat foods with these compounds is low, as, although their absorption is high in their free form, our detoxification system is in charge of eliminating them, and does so very effectively. This, however, does not mean that they do not have favorable or preventive effects against different pathologies. Therefore, based on these data, the intake of foods rich in polyphenols could be advised in order to prevent diseases and as concomitant treatments accompanying pharmacological drugs. However, the usage of high doses of the isolated compounds in humans have not yet been indicated, and therefore further clinical trials are necessary to specify recommended doses.

## Figures and Tables

**Figure 1 biomedicines-09-01074-f001:**
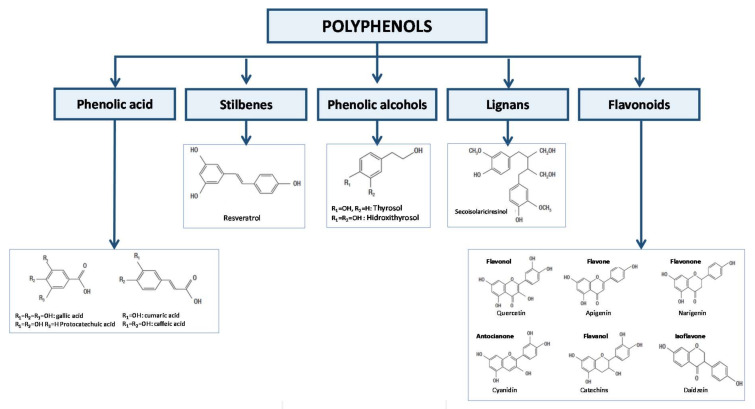
Classification of polyphenols.

**Figure 2 biomedicines-09-01074-f002:**
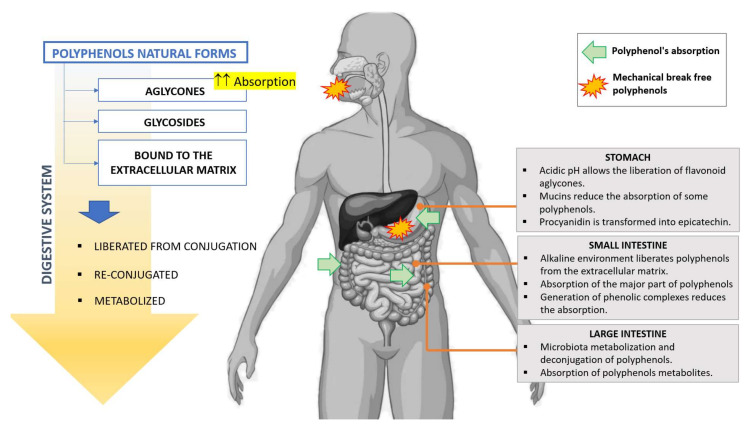
Absorption and metabolism of polyphenols.

**Figure 3 biomedicines-09-01074-f003:**
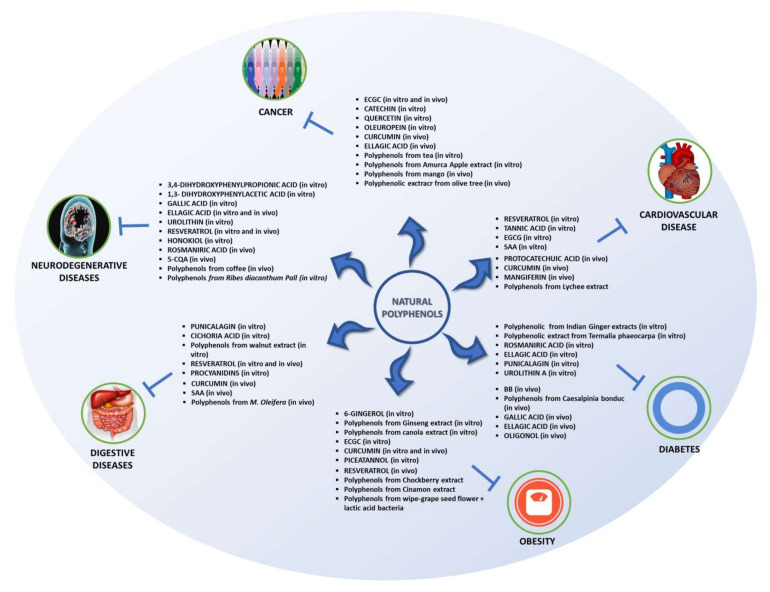
Effects of polyphenols found in animal models and in vitro studies.

**Figure 4 biomedicines-09-01074-f004:**
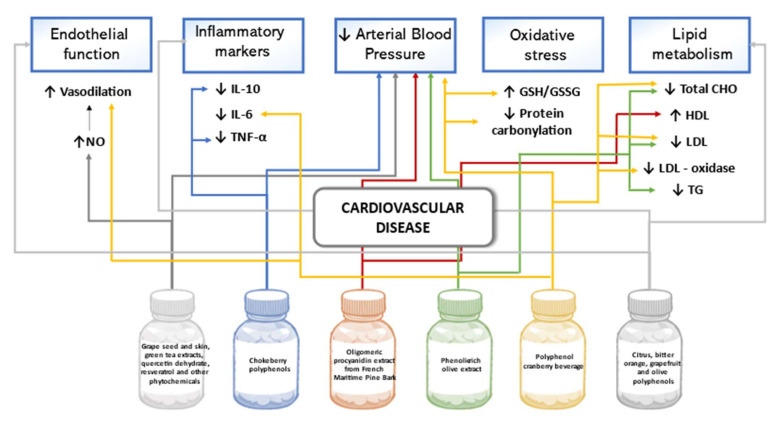
Effects of polyphenols on cardiovascular disease in human studies.

**Figure 5 biomedicines-09-01074-f005:**
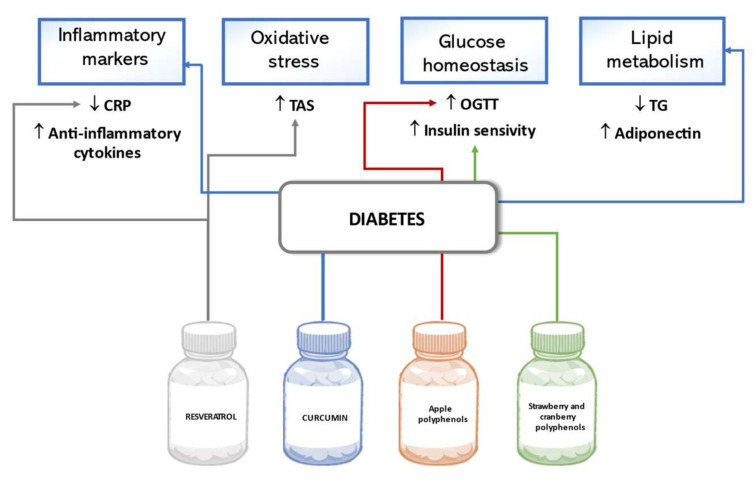
Effects of polyphenols on diabetes in human studies.

**Figure 6 biomedicines-09-01074-f006:**
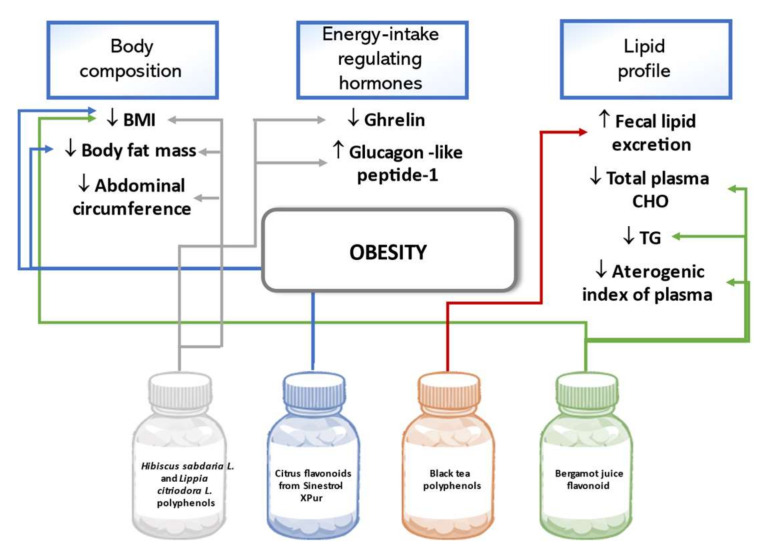
Effects of polyphenols on obesity in human studies.

**Figure 7 biomedicines-09-01074-f007:**
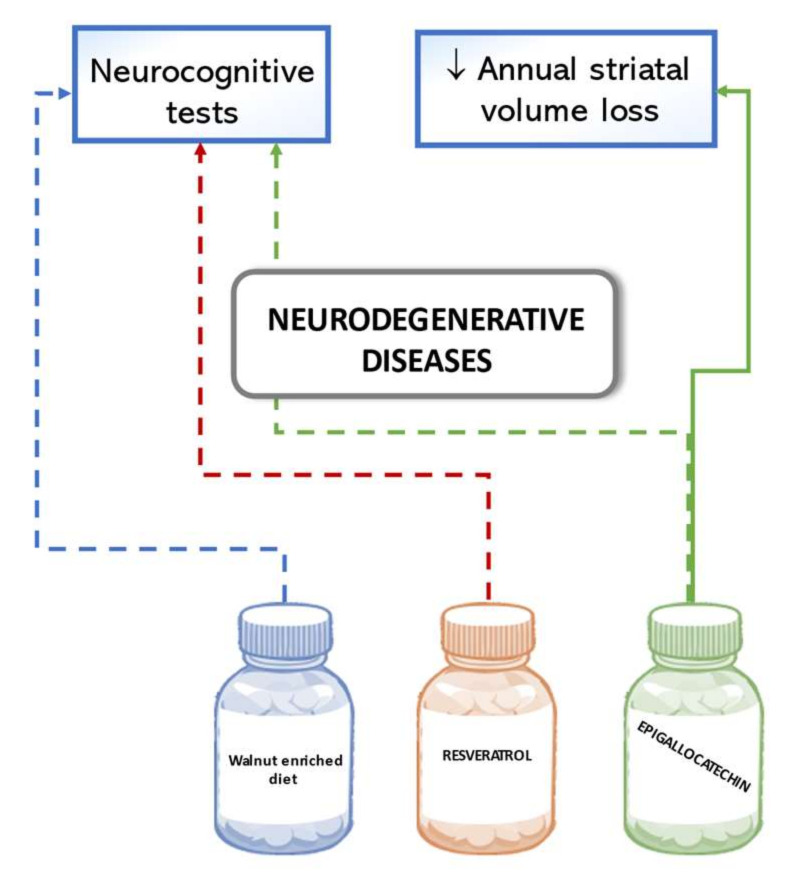
Effects of polyphenols on neurodegenerative diseases in human studies. Dashed lines indicate a lack of significant results.

**Figure 8 biomedicines-09-01074-f008:**
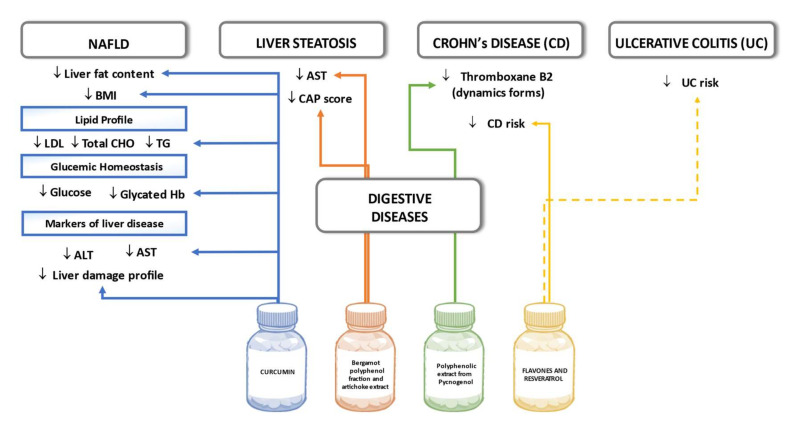
Effects of polyphenols on digestive diseases in human studies. Dashed lines indicate lack of significant results.

**Figure 9 biomedicines-09-01074-f009:**
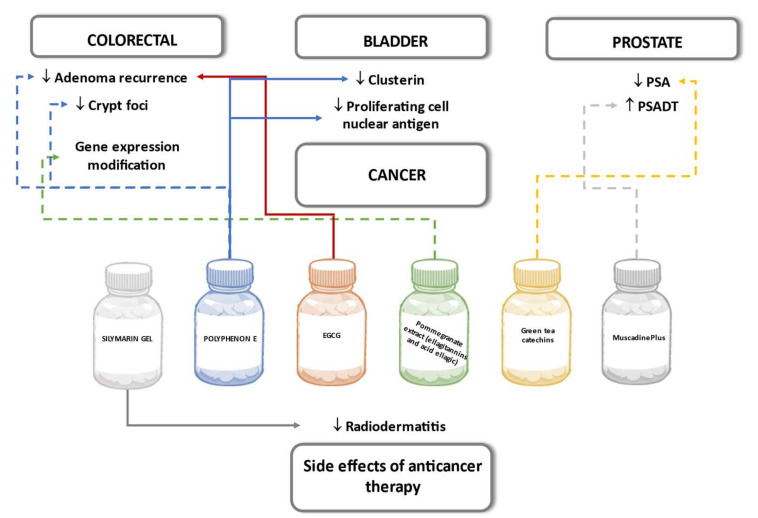
Effects of polyphenols on cancer in human studies. Dashed lines indicate a lack of significant results.

**Figure 10 biomedicines-09-01074-f010:**
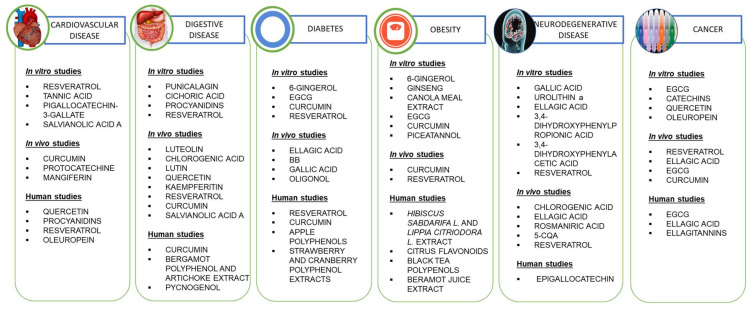
Summary of the effects of polyphenols in animal models, human studies, and in vitro.

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
