# Peer review of "Pharmacological Properties of Polyphenols: Bioavailability, Mechanisms of Action, and Biological Effects in In Vitro Studies, Animal Models, and Humans"

_biomedicines, 2021, doi:10.3390/biomedicines9081074_

Round 1

Reviewer 1 Report

In this descriptive review the authors have summarised available data from the scientific literature about “Pharmacological Properties of Polyphenols:

Bioavailability, Mechanisms of Action, and Biological Effects in Studies in

vitro, in Animal Models and Humans”

The authors conducted an in-depth analysis of bioavailability and pharmacological effects on human health.

The paper is well written and interesting to read, however, I see the following major issues that should be resolved before publishing this paper:

Major

  1. The different paragraphs present all the relevant studies with references but paragraphs 2.1, 2.2, and 2.3 should be revised by adding more practical references. Such as

Ariza, M. T., Reboredo-Rodríguez, P., Cervantes, L., Soria, C., Martínez-Ferri, E., González-Barreiro, C., ... & Simal-Gándara, J. (2018). Bioaccessibility and potential bioavailability of phenolic compounds from achenes as a new target for strawberry breeding programs. Food chemistry248, 155-165.

Feliciano, R. P., Mills, C. E., Istas, G., Heiss, C., & Rodriguez-Mateos, A. (2017). Absorption, metabolism, and excretion of cranberry (poly) phenols in humans: a dose-response study and assessment of inter-individual variability. Nutrients9(3), 268.

  1. Plagiarism has been detected on Page -7,line-283

Xu, L., He, S., Yin, P., Li, D., Mei, C., Yu, X., ... & Liu, F. (2016). Punicalagin induces Nrf2 translocation and HO-1 expression via PI3K/Akt, protecting rat intestinal epithelial cells from oxidative stress. International journal of hyperthermia32(5), 465-473.

https://www.readcube.com/articles/10.3109%2F02656736.2016.1155762

  1. To make the reader move more easily through the text, I suggest the authors go to new lines each time the dissertation needs, thus separating the issues.

  1. It would be advisable to add a methodology section, indicating the user databases, the search terms, etc.
  2. Authors should follow the author's guidelines of the journal and add keywords in abstract.

Minor

  1. Punctuation and grammar problems throughout the manuscript
  2. Page 6 line 221, remove in vivo studies under the invitro section
  3. Page 7 line 280 include cell line
  4. To reader comfort, enlarge the text of figure- 3 and 4
  5. Remove space lines 99 and 505
  6. Add a reference on page 3 line 121

Author Response

In this descriptive review the authors have summarised available data from the scientific literature about “Pharmacological Properties of Polyphenols:

Bioavailability, Mechanisms of Action, and Biological Effects in Studies in

vitro, in Animal Models and Humans”

The authors conducted an in-depth analysis of bioavailability and pharmacological effects on human health.

The paper is well written and interesting to read, however, I see the following major issues that should be resolved before publishing this paper:

Major

The different paragraphs present all the relevant studies with references but paragraphs 2.1, 2.2, and 2.3 should be revised by adding more practical references. Such as

Ariza, M. T., Reboredo-Rodríguez, P., Cervantes, L., Soria, C., Martínez-Ferri, E., González-Barreiro, C., ... & Simal-Gándara, J. (2018). Bioaccessibility and potential bioavailability of phenolic compounds from achenes as a new target for strawberry breeding programs. Food chemistry, 248, 155-165.

Feliciano, R. P., Mills, C. E., Istas, G., Heiss, C., & Rodriguez-Mateos, A. (2017). Absorption, metabolism, and excretion of cranberry (poly) phenols in humans: a dose-response study and assessment of inter-individual variability. Nutrients, 9(3), 268.

Plagiarism has been detected on Page -7,line-283

Xu, L., He, S., Yin, P., Li, D., Mei, C., Yu, X., ... & Liu, F. (2016). Punicalagin induces Nrf2 translocation and HO-1 expression via PI3K/Akt, protecting rat intestinal epithelial cells from oxidative stress. International journal of hyperthermia, 32(5), 465-473.

https://www.readcube.com/articles/10.3109%2F02656736.2016.1155762

Thank you for your suggestions. We have added the outlined articles in the respective paragraphs and the sentence in line 268 has been modified with the corresponding reference added.

To make the reader move more easily through the text, I suggest the authors go to new lines each time the dissertation needs, thus separating the issues.

Thank you for your suggestion. Paragraphs discussion different issues have been separated.   

It would be advisable to add a methodology section, indicating the user databases, the search terms, etc.

Thank you for your suggestion. A methodology section has been made which will be available as supplementary material. If the editors consider that this section would be better in the text on our part we agree.

Authors should follow the author's guidelines of the journal and add keywords in abstract.

Thank you for your observation. Keywords in the abstract have been added.

Minor

Punctuation and grammar problems throughout the manuscript

Page 6 line 221, remove in vivo studies under the invitro section

Page 7 line 280 include cell line

To reader comfort, enlarge the text of figure- 3 and 4

Remove space lines 99 and 505

Add a reference on page 3 line 121

Thank you for your observations. The grammar of the article has been revised. Figures 3 and 4 have been enlarged to facilitate reading. The cell line of the study in line 280 can be found in line 279. All studies outlined on page 6 are in vitro studies. The spaces on line 99 and 505 have been removed as per the reviewers recommendation and the reference on page 3 has been added.

Reviewer 2 Report

Well-organized review reporting the knowledge and studies obtained so far regarding the world of polyphenols. 

I suggest a little spell checking especially for spaces and some formatting errors.

lane 243 " which was found to inhibit pancreatic” remove italics form

lane 388 “administratin”  correct the form in administration

lane 359 “curcumin at concentrations of 25,15 mg/kg/day has been found to reduce serum lipid levels” the concentration is correct?

Author Response

Well-organized review reporting the knowledge and studies obtained so far regarding the world of polyphenols. 

I suggest a little spell checking especially for spaces and some formatting errors.

lane 243 " which was found to inhibit pancreatic” remove italics form

  • Thank you for your kind comments and useful observations. The change has been made.

lane 388 “administratin”  correct the form in administration

  • Thank  you for your observation, the correction has been made.

lane 359 “curcumin at concentrations of 25,15 mg/kg/day has been found to reduce serum lipid levels” the concentration is correct?

  • Thank you for your observation, the concentration has been modified.

Round 2

Reviewer 1 Report

The manuscript has been revised in a proper way.